# SutureBot: A Precision Framework & Benchmark For Autonomous End-to-End Suturing

**Jesse Haworth**[*1]**, Juo-Tung Chen**[*1]**, Nigel Nelson**[2]**, Ji Woong Kim**[3]
**Masoud Moghani**[2,4]**, Chelsea Finn**[3]**, Axel Krieger**[1]

[1]Johns Hopkins University [2]NVIDIA
[3]Stanford University [4]University of Toronto
{jhawort2, jchen396, axel}@jhu.edu

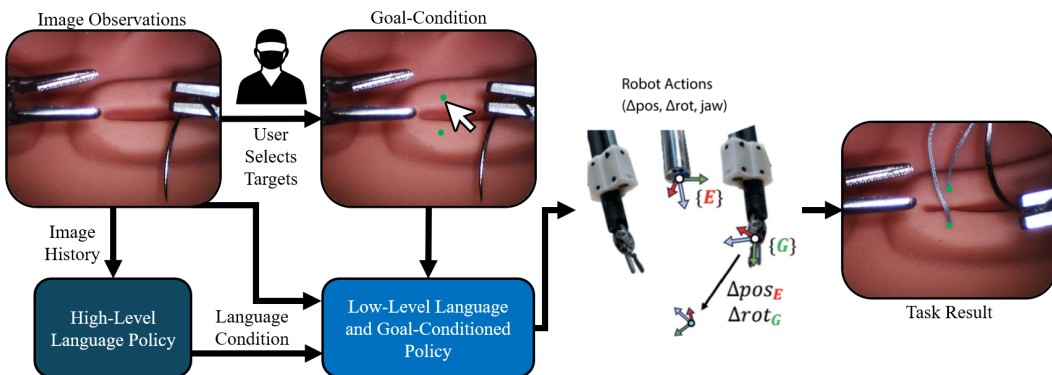

Figure 1: Overview of the precision-conditioned control framework for long-horizon, dexterous surgical tasks. Image observations are processed by a high-level language policy, which selects the current task and generates the associated language condition. The user specifies target needle insertion and exit points via a graphical interface, which is used to generate the goal condition. These inputs, language condition, goal condition, and real-time kinematic data, are then processed by the low-level policy to produce precise, continuous control commands for the robot.

## Abstract

Robotic suturing is a prototypical long-horizon dexterous manipulation task, requiring coordinated needle grasping, precise tissue penetration, and secure knot tying. Despite numerous efforts toward end-to-end autonomy, a fully autonomous suturing pipeline has yet to be demonstrated on physical hardware. We introduce SutureBot: an autonomous suturing benchmark on the da Vinci Research Kit (dVRK), spanning needle pickup, tissue insertion, and knot tying. To ensure repeatability, we release a high-fidelity dataset comprising 1,890 suturing demonstrations. Furthermore, we propose a goal-conditioned framework that explicitly optimizes insertion-point precision, improving targeting accuracy by 59%-74% over a task-only baseline. To establish this task as a benchmark for dexterous imitation learning, we evaluate state-of-the-art vision-language-action (VLA) models, including $\pi_0$, GR00T N1, OpenVLA-OFT, and multitask ACT, each augmented with a high-level task-prediction policy. Autonomous suturing is a key milestone toward achieving robotic autonomy in surgery. These contributions support reproducible evaluation and development of precision-focused, long-horizon dexterous manipulation policies necessary for end-to-end suturing. Dataset is available at: Hugging Face.

39th Conference on Neural Information Processing Systems (NeurIPS 2025) Track on Datasets and Benchmarks.

# 1 Introduction & Related Work

Robotic systems have increasingly demonstrated their potential in enhancing precision, reducing procedural variability, and automating complex tasks across diverse domains, including manufacturing, domestic environments, and healthcare. Within these fields, highly dexterous tasks and generalizable automation remain particularly challenging. Robotic suturing stands out as a paradigmatic example due to its stringent requirements for precision, dexterity, and adaptability to deformation and manipulation uncertainties. Mastering autonomous suturing is a key milestone before automating more complex procedures.

Clinical platforms such as the Da Vinci (Intuitive Surgical, Sunnyvale CA) have shown substantial utility in robotic surgery, offering precise control and enhanced dexterity. However, they rely on continuous surgeon input and face limitations including operator fatigue, human error, and variability in outcomes. The da Vinci Research Kit (dVRK) [11], a widely used research variant, inherits the core capabilities of the clinical system, high-precision control and an intuitive teleoperation interface, while providing a reproducible platform for academic research and experimentation.

Various control methodologies have been explored to achieve differing levels of robotic autonomy in suturing. Hybrid approaches combine motion planning, computer vision, mechanical guides, and predictive modeling. The Smart Tissue Autonomous Robot (STAR) system autonomously executed precise suture placements for small-bowel anastomosis under surgeon supervision, leveraging advanced computer vision strategies and a specialized suturing tool [26]. Suture Needle Angular Positioner (SNAP) [28] and Suture Throws Including Thread Coordination and Handoffs (STITCH) [9] have also employed mechanical guides and sequential convex optimization for accurate suture throws. Knoll et al. [15] utilized scaffolded learning to achieve knot tying for suturing on a real robot. Although these approaches achieve high precision, they often struggle with generalization and error recovery, and have yet to be demonstrated on an end-to-end suturing procedure. Model Predictive Control (MPC) represents another prominent approach, wherein task-specific models optimize robot actions at each timestep. MPC has successfully demonstrated autonomous suture placement on the dVRK [19]. However, MPC often lacks the flexibility to adapt to unpredictable tissue interactions without extensive modeling and has not been used for needle pickup or knot tying during suturing.

Imitation Learning (IL), alternatively, has gained attention due to its ability to learn tasks directly from human demonstrations, offering robust recovery and adaptability. Low-level IL policies target discrete tasks: ACT learns compact action chunks via a transformer backbone to mitigate compounding errors in fine motions [37]; $\pi_0$ leverages a pretrained vision-language model with a flow-matching action expert to generate precise continuous actions [3]. Research into learning individual suturing tasks has led to progress in areas such as needle lifting and handling [16, 32], handover [5], extraction [31], and knot tying [12]. While each of these subtasks has seen successful demonstrations, executing the complete suturing process autonomously continues to be an unsolved problem. These policies achieve high success rates on individual steps but do not address long-horizon sequencing or the precision required for suturing.

For long-horizon coordination, high-level hierarchical frameworks have been proposed. SRT-H uses language-conditioned low-level policies sequenced by a high-level policy to complete ex vivo cholecystectomy procedures [13], but tasks are notably less-dexterous than those needed for suturing. SurgicAI introduces a language-conditioned planner to orchestrate grasp, insert, and handoff tasks and benchmarks multiple IL and RL methods on end-to-end suturing with a 50% success rate [33], however this was only demonstrated in simulation. To address long-horizon coordination in generalist robotic policies, $\pi_{0.5}$ extends $\pi_0$ by using multi-modal data and co-training to achieve long-horizon behaviors, such as laundry folding and box assembly, with robustness to disturbances [10]. However, $\pi_{0.5}$ and recent generalist IL policies that demonstrate the advanced multi-task and long-horizon capabilities required in end-to-end suturing, are trained on 1 million+ trajectories of generalist robotic tasks [6, 3, 20, 14].

Yet, autonomous suturing lacks this quantity of demonstration data to fully realize the recent advances in IL architectures and pretraining. Current publicly available datasets comprised of tabletop tasks using the dVRK system are on the magnitude of a few hundred trajectories [24, 34]. However, datasets specific to autonomous end-to-end suturing total less than 200 trajectories when combined [8, 33]. This data availability greatly limits advancements in solving this canonical surgical task in the real world. As a result, full end-to-end suturing has yet to be demonstrated outside of simulation.

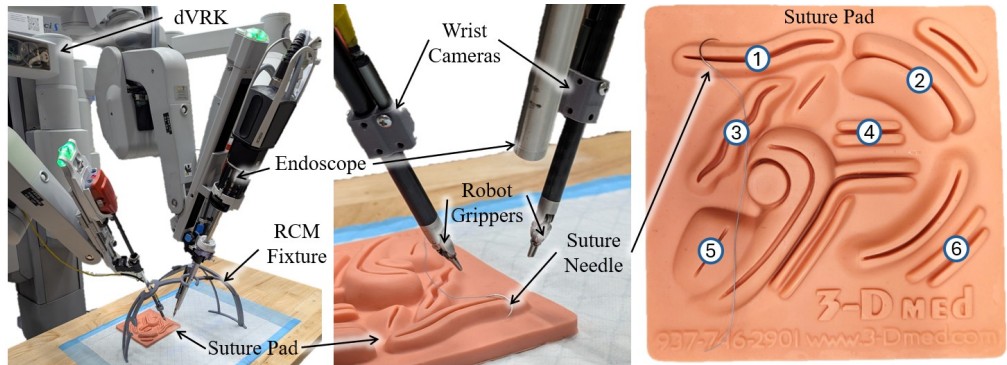

Figure 2: Experimental setup showing the Da Vinci Research Kit (dVRK), remote center of motion fixture, and suture pad. The robot, suture pad, and task utilized for data collection was selected to allow others to reproduce the benchmark. All data is collected on wound one of the suture pad, while wounds two through six are used for generalization testing.

Additionally, there are no established benchmark for the surgical robotics field to track progress in this important autonomous task, nor are there reproducible metrics to assess the level of precision beyond the traditional coarse measure of task completion, which is crucial for downstream clinical significance. To address these gaps and build a foundation for future research, we present a precision-focused IL approach for end-to-end suturing and introduce:

- **A new dexterous benchmark**, featuring a long-horizon suture task for evaluating IL policies in a surgical environment.
- **The largest public real-world suturing dataset**, comprising 1890 high-fidelity dVRK demonstrations for reproducible research.
- **A goal-conditioned IL framework** that enables learned policies to achieve precision-targeted insertion outcomes.
- **A comprehensive evaluation** of state-of-the-art VLA models on our benchmark, establishing a performance baseline for future research.

## 2 Methods

### 2.1 Dataset and Task Description

**System Setup**   Our data-collection setup is shown in Fig. 2. We use the da Vinci Research Kit (dVRK) Si version [35], a widely available research variant of the clinical Da Vinci system. A Soft Tissue Suture Pad (3-D Med, OH) serves as the task surface, and we use a green braided polyester suture (3-0 Ethibond, Ethicon, NJ). All sutures are performed on the region identified in Fig. 2.

A 3D-printed trocar cage maintains the fixed remote centers of motion (RCMs); the corresponding STL file is included in the dataset. We equip the dVRK with DeBakey forceps on the left arm and a large needle driver on the right. Wrist cameras (5.5 mm borescope, Takmly, China) are mounted 35 mm from each wrist via 3D-printed fixtures (STL file in dataset). An absorbent pad beneath the suture pad and RCM fixture provides a consistent background. During collection, we record images at 30 Hz synchronized with robot kinematics.

**Task Description**   Suturing is a fundamental surgical task involving the precise placement of a needle and thread to join tissue, promote healing, and achieve hemostasis. One common approach is the interrupted stitch, where the needle is passed through both sides of the tissue to be connected, the suture is tied securely, and the excess thread is trimmed.

We decompose suturing into three tasks, following the task breakdown used in SRT [12]. Examples are shown in Fig. 3. **Needle pickup** begins with both grippers positioned above the wound and the needle resting on the pad. The left gripper grasps the needle near its tip, then hands it off to the right

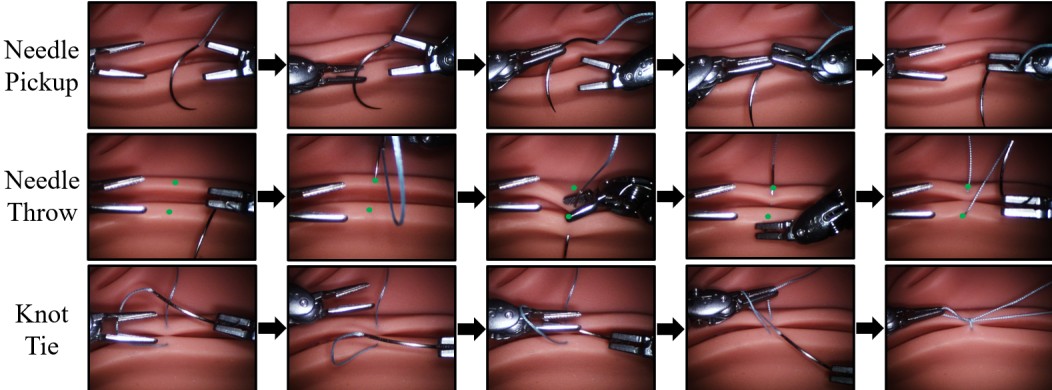

Figure 3: The suturing procedure is broken into three tasks, needle pickup, needle throw, and knot tie. These task discretizations were then utilized for data collection, policy training, and evaluation.

gripper, which grasps near the base, with the curve of the needle oriented away from the endoscope camera. **Needle throw** uses the right gripper to drive the needle through the back wall of the wound, rotate it, and pass it through the front wall. Once sufficient suture emerges, the right gripper releases the needle, repositions to grasp the protruding suture, and pulls it straight through. Once the needle is free of the suture pad, it is pulled straight up, drawing suture material through before returning to home. **Knot tying** is performed by the right gripper wrapping the suture clockwise around the left gripper. The left gripper then opens slightly, grasps the loose end of the suture and pulls to the left while the right gripper pulls the needle to the right to tighten the knot.

**Data Collection and Dataset Composition**    We collect demonstrations for three tasks that comprise the suturing procedure, *needle pickup*, *needle throw*, and *knot tying*, as well as corresponding *recovery demonstrations*, where the task begins from a failure state and proceeds to successful completion. These recovery demonstrations are inspired by the DAgger [25] framework, where an initial policy trained on expert demonstrations is deployed to identify common failure modes. We then collect additional demonstrations that start from these failure states, showing how to recover and complete the task. This approach increases the diversity of the training data and helps the policy generalize beyond ideal conditions. It also improves robustness by explicitly teaching the model how to recover from suboptimal states that are likely to occur during real-world deployment.

After data collection, we manually annotate each needle throw demonstration with insertion and exit points on the final frame using a GUI. These annotations are stored as x and y image coordinates in CSV files in each demonstration episode and are later used as goal conditions for training and evaluation.

In total, we collected 1,890 demonstrations, including 454 recovery examples. This dataset comprises 628 demonstrations for needle pickup (148 recoveries), 310 for needle throw (96 recoveries), and 952 for knot tying (210 recoveries). All demonstrations were collected using the standard dVRK teleoperation console, allowing fine-grained manual control of both arms.

Each demonstration includes synchronized visual and kinematic data. We record robot kinematics in structured CSV files at each timestamp. These logs include 6-DOF Cartesian poses (position and quaternion) of both end-effectors, measured jaw opening angles, desired Cartesian poses and joint angles, and the pose of each remote center of motion (RCM) frame. Additionally, we capture RGB images from the stereo endoscope and two wrist-mounted cameras. Wrist cameras record at a resolution of $640 \times 480$ at 30 Hz, and the stereo endoscope records at $960 \times 540$ at the same frame rate.

To improve policy robustness and generalization, we introduce variation across demonstrations. This includes differences in robot joint configurations, RCM positions within the fixture, the placement and orientation of the suture pad, the initial pose of the needle, and slight perturbations to the wrist camera mounts. These variations ensure a diverse set of trajectories and visual scenes across the dataset.

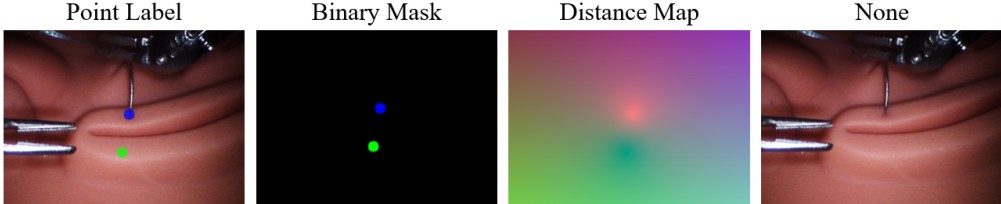

Figure 4: To enable a policy to approach targeted points, we utilize goal conditions generated from target points, which serve as inputs to the model during training and inference. We evaluate three types of goal conditions; point labels on the endoscope image, an additional image input with masks, and an additional image input with a distance map.

## 2.2 Policy Architecture

**Architecture Overview**  We adopted a hierarchical architecture similar to [13, 29, 30], which is shown in Fig. 1. A high-level policy based on the Swin Transformer [18] encodes the visual observations into tokens, that are processed by a transformer decoder to generate language instructions. The low-level policy receives this language instruction, along with the latest wrist and endoscope images, and the goal conditions, and outputs a chunk of relative robot actions.

For the low-level policy, we compare three state-of-the-art vision-language-action (VLA) models, $\pi_0$ [3], GR00T N1 (GR00T) [20], and OpenVLA-OFT (OpenVLA) [14]. These models leverage vision-language model (VLM) backbones pretrained on internet-scale datasets and have demonstrated strong performance on a wide range of general-purpose manipulation tasks. The $\pi_0$ and GR00T N1 models represent strong VLA generalist policies, whose flow-matching action prediction heads and foundational pretraining have been demonstrated to enhance downstream finetuning tasks [3, 20]. OpenVLA-OFT differs from these approaches by leveraging parallel decoding for action prediction, which when coupled with L1 regression and FiLM conditioning [22] earn it SOTA performance on the LIBERO simulation benchmark [17]. In addition, we include a language-conditioned Action Chunking Transformer (ACT) as a baseline. Unlike the VLA models, ACT does not rely on a pretrained vision-language model (VLM) backbone, and therefore serves as a non-VLA reference point for comparison. The use of language conditioning allows ACT to be trained on multiple tasks within a single model, in contrast to prior work such as SRT [12], which required training separate models for each task. This multitask formulation facilitates smoother transitions between tasks and supports a unified low-level policy for the entire suturing procedure. We attempted to apply this language conditioned approach to Diffusion Policy (DP) [4], however, we encountered challenges in achieving reasonable performance with our implementation, leading to its exclusion from the final comparison.

Further implementation details are provided in the appendix.

**Training**  We reserve an evaluation set of two demonstrations per task and its corresponding recovery (12 demos total). Models trained with L1 regression, ACT and OpenVLA-OFT, are trained for at least 10,000 steps. However, models trained with MSE, $\pi_0$ and GR00T N1, often require much fewer steps due to a propensity to overfit. Each final checkpoint is selected according to the lowest evaluation loss achieved before overfitting is observed. Additional training hyperparameters are detailed in the appendix. All training is conducted on an NVIDIA DGX A100 system with 8x A100 80 GB GPUs.

## 2.3 Goal Representations

Goal conditioning has been established in prior works, such as RoboPoint [36] and more recently with AimBot [7], which are used to boost task success rates, but have yet to be demonstrated for measured precision control of IL policies. We explore three goal condition formats to guide needle placement (Fig. 4). **Point labels:** The endoscope image is overlaid with an opaque blue pixel at the insertion point and an opaque green pixel at the exit point. **Binary masks:** A three-channel image, where channel 2 represents the insertion mask, channel 3 the exit mask, and channel 1 is all zeros. **Distance maps:** A three-channel image, where the first two channels encode normalized pixel-wise offset vectors $(dx, dy)$ pointing toward the insertion point, and the third channel is a scalar heatmap

| UV Mark Wound | Endoscope View | Select Points | Execute Task | Measure Error |

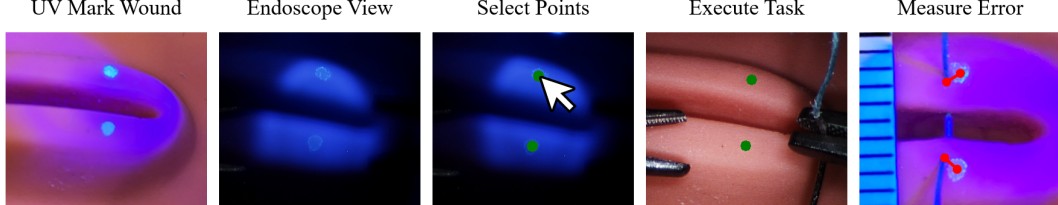

Figure 5: Ultraviolet (UV) marks were utilized to measure the accuracy of the policy. After marking the wound, the target points were selected in the endoscope view using a UV light. During execution, the UV light was off and the marks were invisible. After the task, the UV light was turned on and the distance from the suture to the mark was measured.

with intensity highest at the insertion point and lowest at the exit point. Binary masks and distance maps are passed to the low-level policy as separate inputs, while point labels directly modify the endoscope image. **No Goal:** Policies without explicit goal conditions rely solely on the distribution of insertion points in the training data. We include this baseline to quantify the error attributable to this distribution, distinguishing it from policies that explicitly learn to reach specified goals.

# 3 Evaluation and Results

Our evaluation aims to address four key questions:

- Which goal–conditioning representation yields the highest precision for suturing?
- How do state-of-the-art IL models perform on the SutureBot benchmark?
- How does pretraining affect policy performance?
- How well does this approach generalize to previously unseen scenarios?

## 3.1 Metrics

**Needle Pickup**: A pickup trial was considered successful if the left gripper first grasps the needle and the right gripper subsequently secures it. Trials exceeding 120 s are marked as failures.

**Needle Throw**: While needle throw and pull through are trained as one task, for evaluation it was broken up into two sub-tasks, throw and pull through. A throw trial succeeds if the needle penetrates the back wall and then the front wall of the wound within 120 s. If the model was able to pull the needle free from the tissue and return to the home position within 60 s it was considered a successful pull through sub-task.

**Knot Tie**: A knot-tying trial was successful if the right gripper wraps the suture clockwise around the left gripper, and the left gripper grasps and tightly pulls the loose end through the loop within 120 s.

**Insertion and Exit Error**: To evaluate precision, we used invisible ultraviolet (UV) markers. Before execution, the target insertion and exit points on the pad were marked using a UV pen as shown in Fig. 5. The suture pad was then illuminated with a UV light under the endoscope camera, allowing the digital target points to be selected. These target points were used to generate the goal condition which was passed to the policy. After completion, the pad was re-illuminated with a UV light and ImageJ [27] was used to measure the Euclidean distance between the intended UV marks and the actual suture insertion/exit points which define the insertion and exit errors. If the throw task failed, but the policy completed at least one puncture, the measurement for that puncture is still included.

**Procedure Time**: Total time reported for each procedure was calculated by adding the recorded time from all successful tasks in the procedure to the maximum time for each failed task as defined above. Time is marked as not available (NA) if the policy failed to complete any tasks.

## 3.2 Ablation Studies

We evaluated each policy in a fixed robot configuration with consistent suture pad placement and varying needle position. All models were evaluated on a dual NVIDIA RTX 4090 workstation. Each

policy executes ten full suture procedures, with individual task successes and errors recorded. If a task fails, the system is manually reset before the next task.

**Goal Condition Representation**  We fine-tune $\pi_0$ and train multitask ACT using four goal condition formats: point labels, binary masks, distance maps, and no conditioning. Table 1 summarizes their performance on the throw task along with precision metrics. All goal-conditioned variants appear to improve accuracy compared to the no-goal baselines, with point labels achieving the lowest average error of 1.3 mm for ACT and 1.0 mm for $\pi_0$.

For statistical analyses, we used the Real Statistics Resource Pack in Microsoft Excel (Release 9.1.1; 2024, Charles Zaiontz). We compared the Point Label goal conditioning method against Distance Map, Mask, and no goal separately for the ACT and $\pi_0$ policies. We used non-parametric Mann-Whitney U tests for accuracy and Brown-Forsythe tests for precision, both with a Bonferroni-corrected threshold ($\alpha = 0.0167$). For ACT, Point Label was significantly more accurate than Distance Map ($p = 0.010$) and Mask ($p = 0.009$), and more precise than Mask ($p = 0.009$) and no goal ($p = 0.013$). For $\pi_0$, point label was significantly more accurate than Distance Map ($p = 0.010$) and no goal ($p = 0.002$), and more precise than no goal ($p = 0.007$). Overall, Point Label demonstrated the most consistent benefits for improving policy accuracy and precision, especially for ACT. As a result, all subsequent policy evaluations use the point-label representation.

Table 1: Success rates and precision results for different goal conditions on the suturing procedure. Error results reported in Avg±Std mm.

| Policy | Throw | Pull Through | Insertion Error (mm) | Exit Error (mm) | Time (sec) |
|---|---|---|---|---|---|
| ACT + Point Label | 9/10 | 7/10 | 1.3±0.9 | 2.0±1.3 | 70±17 |
| ACT + Distance Map | 8/10 | 8/10 | 2.6±1.5 | 2.2±1.8 | 76±16 |
| ACT + Mask | 10/10 | 4/10 | 2.9±1.7 | 3.0±1.0 | 91±19 |
| ACT (no goal) | 10/10 | 9/10 | 3.2±2.2 | 3.6±1.8 | 72±14 |
| $\pi_0$ + Point Label | 6/10 | 2/10 | 1.0±1.3 | 2.4±1.6 | 129±48 |
| $\pi_0$ + Distance Map | 8/10 | 1/10 | 2.1±1.1 | 2.3±0.9 | 124±8 |
| $\pi_0$ + Mask | 6/10 | 4/10 | 1.8±1.2 | 2.1±1.2 | 134±27 |
| $\pi_0$ (no goal) | 8/10 | 3/10 | 3.9±2.5 | 3.7±2.5 | 130±30 |

**Low-Level Policy Comparison**  We finetune $\pi_0$, GR00T N1, OpenVLA-OFT, and train multitask ACT on the SutureBot dataset and evaluate their capabilities as low-level policies. Table 2 reports the success rates and mean insertion/exit errors for each model. For individual task completion, ACT performs the best, followed by $\pi_0$. ACT also completed 3/10 sutures end-to-end, with no manual intervention between tasks, while also having the best insertion error with an average of 1.5±0.8 mm followed closely by $\pi_0$ with 1.9±1.0 mm. We performed statistical analyses using a 4x2 Chi-squared test, which showed a highly significant overall difference ($p = 8.4e - 14$). Post-hoc analysis involved pairwise one-tailed Fisher's Exact tests comparing the best model (ACT) against the others, using a Bonferroni-corrected threshold ($\alpha = 0.0167$). Results showed ACT performed significantly better than GR00T N1 ($p = 8.8e - 9$), and OpenVLA-OFT ($p = 2.1e - 12$), but not significantly better than $\pi_0$ ($p = 0.018$).

Table 2: Success rates and precision results of the evaluated models on the suturing procedure. Error and time results reported in Avg±Std.

| Policy | Pickup | Throw | Pull Through | Knot Tie | Insertion Error (mm) | Exit Error (mm) | Time (sec) | End-to-End |
|---|---|---|---|---|---|---|---|---|
| ACT | 9/10 | 8/10 | 4/10 | 9/10 | 1.5±0.8 | 2.6±1.2 | 182±58 | 3/10 |
| $\pi_0$ | 7/10 | 7/10 | 3/10 | 4/10 | 1.9±1.0 | 3.2±2.3 | 348±45 | 0/10 |
| GR00T | 1/10 | 2/10 | 1/10 | 1/10 | 2.3±1.2 | 2.9±0.6 | 388±67 | 0/10 |
| OpenVLA | 0/10 | 0/10 | 0/10 | 0/10 | NA±NA | 2.8±NA | NA | 0/10 |

**High-Level Policy and Pretraining Evaluation** The high-level policy achieved an F1 score of 0.92 and accuracy of 88.73% for task prediction during offline validation. It also achieved 100% F1 score and accuracy in detecting task transitions, indicating reliable classification of the overall procedure into discrete subtasks. A more detailed confusion matrix is shown in the appendix. To further assess its effectiveness, we conduct an oracle comparison in which a human operator manually provides language conditions to the low-level policy, replacing the high-level policy for direct evaluation ($\pi_0$ Oracle and ACT Oracle).

For pretraining evaluation, we compare three configurations of the best-performing VLA policy, $\pi_0$: (1) the standard $\pi_0$ checkpoint used in other fine-tuning evaluations, (2) a variant post-trained on 20,000 predominantly cholecystectomy trajectories from SRT-H ($\pi_0$ Chole) [13], and (3) a version initialized from scratch with the backbone VLM from a standard PaliGemma checkpoint [2] ($\pi_0$ Scratch). All models were then fine-tuned on the SutureBot dataset prior to evaluation. Results, summarized in Table 3, indicate that $\pi_0$ and $\pi_0$ Chole achieved comparable task success rates, with $\pi_0$ showing a slight advantage in insertion error. The oracle results closely matched those of $\pi_0$, suggesting the high-level policy performed comparably to a human operator in directing the low-level policy.

Table 3: Success rates and precision results of $\pi_0$ pretraining checkpoints on the suturing procedure along with an oracle evaluation of the high-level policy. Error and time results reported in Avg±Std.

| Policy | Pickup | Throw | Pull Through | Knot Tie | Insertion Error (mm) | Exit Error (mm) | Time (sec) | End-to-End |
|--------|--------|-------|--------------|----------|---------------------|-----------------|------------|------------|
| ACT | 9/10 | 8/10 | 4/10 | 9/10 | 1.5±0.8 | 2.6±1.2 | 182±58 | 3/10 |
| ACT Oracle | 7/10 | 7/10 | 5/10 | 10/10 | 1.8±0.8 | 2.6±1.1 | 207±48 | 2/10 |
| $\pi_0$ | 7/10 | 7/10 | 3/10 | 4/10 | 1.9±1.0 | 3.2±2.3 | 348±39 | 0/10 |
| $\pi_0$ Oracle | 3/10 | 9/10 | 3/10 | 7/10 | 1.3±0.4 | 3.1±2.1 | 313±51 | 0/10 |
| $\pi_0$ Chole | 4/10 | 4/10 | 0/10 | 5/10 | 2.2±0.7 | 3.5±1.6 | 342±66 | 0/10 |
| $\pi_0$ Scratch | 6/10 | 2/10 | 2/10 | 1/10 | 3.7±3.6 | 3.9±1.0 | 364±42 | 0/10 |

### 3.3 Generalization

We evaluated the best and next best-performing policy's ability to generalize by testing on wound geometry not included in the training data. Fig. 2 shows the 3-D Med suture pad with wound types one through six labeled. Wound one was used in the training data while wounds two through six were excluded. We then evaluated the policies under modified lighting conditions by using an darker external lamp for lighting the scene from the side instead of the direct bright endoscope light. Lastly, we tested the policies using a different tool set than was in the training data by switching the left DeBakey forceps and right Large Needle Driver. Table 4 summarizes the policy's performance results on all three generalization conditions. The results of $\pi_0$ on unseen wound types are extremely comparable to those on the trained wound, while ACT has a noticeable performance drop. Success rates drop further with the new lighting and tool configurations for both ACT and $\pi_0$.

## 4 Discussion

**Goal Condition Representation** Evaluating multiple goal conditioning methods, we find that overlaying the original endoscope image with opaque point labels yields the lowest insertion error and variance for the needle throw sub-task, though exit error remains comparable across non-baseline methods. This may result from a shared limitation: all models lack historical context, leading to uncertainty when the needle is obscured within the tissue. This underscores the challenge of achieving high exit precision, where initial positioning and entry angles significantly constrain the possible exit points.

Interestingly, models trained with point labels tend to align the needle more carefully as they approach the target, often displaying deliberate, hesitant motions during insertion. This suggests better spatial awareness and fine-grained control. In contrast, models conditioned on distance maps or binary masks complete the task more quickly but with reduced accuracy. This discrepancy may arise from

Table 4: Success rates and precision results on unseen wound types. "on (1)" are the results from the wound used during training. "on (2-6)" are the results from wound types not included in the training data. Fig. 2 shows wounds one through six on the suture pad. "Lighting" are the results with alternate lighting from the training set and "Tools" are results with alternate tools from the training set. Error results reported in Avg±Std.

| Scene Change | Pickup | Throw | Pull Through | Knot Tie | Insertion Error (mm) | Exit Error (mm) | Time (sec) | End-to-End |
|---|---|---|---|---|---|---|---|---|
| ACT (1) | 9/10 | 8/10 | 4/10 | 9/10 | 1.5±0.8 | 2.6±1.2 | 182±58 | 3/10 |
| ACT (2-6) | 5/10 | 6/10 | 2/10 | 5/10 | 1.2±0.8 | 2.5±1.3 | 276±95 | 0/10 |
| ACT Lighting | 3/10 | 5/10 | 7/10 | 4/10 | 2.2±0.9 | 2.7±0.9 | 349±53 | 0/10 |
| ACT Tools | 1/10 | 9/10 | 1/10 | 2/10 | 1.1±0.5 | 2.6±0.6 | 327±22 | 0/10 |
| $\pi_0$ on (1) | 7/10 | 7/10 | 3/10 | 4/10 | 1.9±1.0 | 3.2±2.3 | 348±39 | 0/10 |
| $\pi_0$ on (2-6) | 5/10 | 6/10 | 0/10 | 8/10 | 2.0±1.5 | 2.5±1.1 | 293±57 | 0/10 |
| $\pi_0$ Lighting | 0/10 | 5/10 | 0/10 | 2/10 | 2.7±2.2 | 4.1±2.6 | 402±28 | 0/10 |
| $\pi_0$ Tools | 1/10 | 5/10 | 0/10 | 3/10 | 3.1±1.3 | 4.8±1.1 | 383±48 | 0/10 |

the added cognitive load of integrating a separate input (mask or map) with the endoscopic view, whereas point overlays directly embed the goal representation within the task image, providing a more explicit and intuitive target.

**Low-Level Policy Comparison** Of the evaluated low-level policies, ACT achieves the highest task completion rate, as well as suture throw precision matched only by $\pi_0$. We attribute this advantage partially to the dataset being smaller and relatively uniform data which smaller policies like ACT may benefit from. Although policies like $\pi_0$ may be able to leverage pretraining to finetune for this task, the nature of the task and dVRK being significantly different from $\pi_0$'s pretraining potentially limit pretraining advantages. Due to similarities in architecture, the performance advantage that $\pi_0$ offers over GR00T N1 can likely be attributed to the former's pretraining that emphasized bimanual manipulators that more closely resemble the dVRK platform compared to the latter whose pretraining focused on humaniods with more degrees of freedom.

**Pretraining Evaluation** In Table 3, we show that while the baseline model fine-tuned from $\pi_0$'s public checkpoints slightly outperforms other pretrained variants in average task completion, the results do not clearly differentiate in-domain benefits across the various pretraining mixtures. It is noted that the $\pi_0$ from scratch policy only had a small performance drop compared to $\pi_0$ from checkpoint, which supports the idea that this task and embodiment are significantly different from $\pi_0$ pretraining, limiting its benefit. Finally, pretraining did reduce time to convergence, with $\pi_0$ Scratch requiring more training time than the baseline, and $\pi_0$ Chole converging the fastest.

**High Level Policy** To evaluate the impact of high-level policy on overall execution, we compare $\pi_0$ and multitask ACT (which uses the high-level policy) to an oracle variant with human-selected subtasks. As shown in Table 3, the high-level and oracle variants achieve similar success rates, timing, and precision, across both ACT and $\pi_0$, indicating that the high-level policy provides sufficiently accurate subtask predictions. These results suggest that the high-level policy is not a performance bottleneck and enables effective multitask execution in autonomous suturing.

**Generalization** One advantage of IL, particularly vision-language-action (VLA) models like $\pi_0$, is their ability to generalize to unseen environments, a common challenge for traditional model-based approaches. In our experiments, the $\pi_0$ policy performed consistently across wounds with varying thickness and geometry, similar to its performance on the training set wound. ACT had a noticeable drop in performance on the wound set and both policies were significantly worse on the alternate tools and lighting sets. While even with a simple dataset, there is some generalization ability, this highlights the importance of a diverse dataset especially when using IL for surgical applications, where the suturing environment can vary significantly.

# 5 Limitations

This work has several limitations. First, the number of trials per experiment is limited to 10 due to time constraints. While this is sufficient for identifying clear trends, more subtle effects, such as those observed in pretraining evaluations, may benefit from larger sample sizes. High-level policy evaluations were limited to offline and Oracle evaluations. This manuscript focused on evaluating the low-level policies, but future work should include more discrete online evaluations of the high-level policy. While our framework shows some ability to generalize to variations in suture pad wound types, performance drops under light and tool changes and is likely to degrade further under more novel conditions (e.g., different pad materials, phantom blood, real tissue), highlighting the need for more diverse training data to improve generalization. Our pipeline currently relies on manual target-point selection, which will need to be automated for full autonomy, as was demonstrated in [26]. Although our clinical post-training did not significantly improve performance in this work, this remains a promising area for future research, as generalist robotic foundation models [10, 20, 14] have demonstrated strong downstream transfer when aligned with their training domain. Our dataset will also be useful in training and developing these foundation models or pretraining for other dual-armed robots. As a first step towards automating suturing, we focus on simple success and error metrics, however, in future work clinical metrics such as bite depth, tissue trauma, and suture tension will need to be considered to improve clinical relevance. Finally, while this dataset and benchmark was designed to foster end-to-end autonomous suturing advancement, only ACT achieved end-to-end success in 3 trials without human operator input. We aim to address this, as well as the other limitations discussed in future work by investigating alternative architectures, pretraining, and expanding the dataset. Expanding the dataset scale and including more diversity will help determine whether the performance bottleneck is due to dataset size or policy architecture.

# 6 Societal Impact

An estimated 67% of the world's population lacks access to surgical care [1]. Even in countries like the United States, with relatively high surgical access, an aging population is expected to create a shortage of 10,100 to 19,900 surgical specialists by 2036 [23]. Increased autonomy in robotic surgical systems could help address these shortages by expanding the capacity of the existing surgical workforce. Robotic-assisted surgery (RAS) has already been shown to reduce healthcare costs and patient length of stay [21], but current RAS systems offer limited autonomy. Our work aims to support future advances in autonomous systems that could further improve surgical efficiency and outcomes. However, the methods and benchmarks presented in this work also carry the risk of premature exploration of surgical automation without sufficient regard for safety and ethical considerations. We emphasize that this is exploratory research, with performance well below that of expert surgeons. Significant additional work is required to improve accuracy and carefully assess the ethical implications before deployment in clinical settings.

# 7 Conclusion

We have introduced the largest publicly available autonomous, end-to-end suturing dataset and benchmark on the widely used dVRK platform, and demonstrated that current VLAs finetuned on SutureBot can achieve each individual task, but lack the consistency to reliably demonstrate complete end-to-end suturing examples. Additionally, we find that VLAs augmented with goal conditioning can achieve a mean insertion error of 1.0±1.3mm. Our goal-conditioned IL framework provides a 59%-74% improvement in suturing precision over baseline, and our public benchmark and high-fidelity dataset enable reproducible progress in precise, long-horizon, and dexterous manipulation.

## Acknowledgments and Disclosure of Funding

This material is based on work supported by NIH R56EB033807 and ARPA-H AY1AX000023. We would also like to thank NVIDIA for sharing computational resources for training the policies.

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

## A   Technical Appendices and Supplementary Material

Table 5: $\pi_0$ finetuning parameters and variations.

| Pretrained ckpt | $\pi_0$ | PaliGemma | $\pi_0$ | $\pi_0$ Chole |
|---|---|---|---|---|
| Training dataset | SutureBot | SutureBot | Chole | SutureBot |
| Learning rate | 1e-4 | 1e-4 | 1e-4 | 1e-4 |
| Optimizer | AdamW | AdamW | AdamW | AdamW |
| Adam beta1 | 0.9 | 0.9 | 0.9 | 0.9 |
| Adam beta2 | 0.95 | 0.95 | 0.95 | 0.95 |
| Adam epsilon | 1e-8 | 1e-8 | 1e-8 | 1e-8 |
| Weight decay | 0.1 | 0.1 | 0.1 | 0.1 |
| LR scheduler | Cosine | Cosine | Cosine | Cosine |
| Batch size | 128 | 128 | 128 | 128 |
| Gradient steps | 7,500 | 8,000 | 20,000 | 6,000 |
| Warmup steps | 1,000 | 1,000 | 1,000 | 1,000 |
| Full finetune | True | True | True | True |
| Training chunk size | 50 | 50 | 50 | 50 |
| Eval action horizon | 20 | 20 | 20 | 20 |
| Training data FPS | 30 Hz | 30 Hz | 30 Hz | 30 Hz |

Table 6: GR00T N1 finetuning parameters.

| Parameter | Value |
|---|---|
| Learning rate | 1e-5 |
| Optimizer | AdamW |
| Adam beta1 | 0.95 |
| Adam beta2 | 0.999 |
| Adam epsilon | 1e-8 |
| Weight decay | 0.1 |
| LR scheduler | Cosine |
| Batch size | 128 |
| Gradient steps | 3,000 |
| Warmup steps | 200 |
| Tune vision | True |
| Tune projector | True |
| Tune diffusion head | True |
| Tune LLM | False |
| Training chunk size | 16 |
| Eval action horizon | 16 |
| Training data FPS | 15 Hz |

Table 7: OpenVLA-OFT finetuning parameters.

| Parameter | Value |
| --- | --- |
| Learning rate | 5e-5 |
| Optimizer | AdamW |
| Adam beta1 | 0.9 |
| Adam beta2 | 0.999 |
| Adam epsilon | 1e-8 |
| Weight decay | 0.01 |
| LR scheduler | MultiStepLR |
| LR gamma | 0.1 |
| Batch size | 32 |
| Gradient steps | 29,000 |
| Warmup steps | 100 |
| Use LoRA | True |
| LoRA rank | 32 |
| LoRA dropout | 0 |
| Training chunk size | 50 |
| Eval action horizon | 20 |
| Training data FPS | 30 Hz |

Table 8: Multitask ACT training parameters.

| Parameter | Value |
| --- | --- |
| Learning rate | 5e-4 |
| Optimizer | AdamW |
| Adam beta1 | 0.9 |
| Adam beta2 | 0.999 |
| Adam epsilon | 1e-8 |
| Weight decay | 1e-4 |
| LR scheduler | LambdaLR |
| Batch size | 256 |
| Gradient steps | 10,000 |
| Warmup steps | 500 |
| KL weight | 10 |
| Hidden dim | 512 |
| Feedforward dim | 3200 |
| Train chunk size | 60 |
| Eval action horizon | 20 |
| Training data FPS | 30 Hz |
| Use FiLM | True |
| Language encoder | DistilBERT |
| Image Encoder | EfficientNet-B3 |

Table 9: High-Level Policy training parameters.

| Parameter | Value |
|---|---|
| Learning rate | 4e-4 |
| Minimum LR | 1e-5 |
| LR cycle length | 25 epochs |
| Warmup epochs | 5 |
| Batch size | 16 |
| Weight decay | 0.05 |
| Num epochs | 2000 |
| Best val epoch | 282 |
| Validation interval | 10 epochs |
| Save checkpoint interval | 5 epochs |
| Early stopping interval | 300 epochs |
| Seed | 5 |
| Prediction offset | 15 |
| History length | 4 frames |
| History step size | 30 frames |
| Cameras used | left_img_dir |
| Image resolution | $224 \times 224$ |
| Backbone model | Swin-T |
| Init weights | ImageNet |
| Freeze backbone until | none |
| Multitask loss weight | 0.6 |
| Use complex MLP head | True |
| Selected multitasks | dominant_moving_direction |
| Recovery probability | 0.6 |
| Use one-hot subtask labels | True |
| Uniform sampling | True |
| Extra repeated last-frame sampling | True |
| Extra sampling probability | 0.15 |
| Add center crop view | True |
| Use global pooled image features | True |


Figure 6: Confusion matrix for the High Level Policy on Validation dataset

Justification: In the limitations section we discuss the limitations, implication, and future work related to our research.

Guidelines:

- The answer NA means that the paper has no limitation while the answer No means that the paper has limitations, but those are not discussed in the paper.
- The authors are encouraged to create a separate "Limitations" section in their paper.
- The paper should point out any strong assumptions and how robust the results are to violations of these assumptions (e.g., independence assumptions, noiseless settings, model well-specification, asymptotic approximations only holding locally). The authors should reflect on how these assumptions might be violated in practice and what the implications would be.
- The authors should reflect on the scope of the claims made, e.g., if the approach was only tested on a few datasets or with a few runs. In general, empirical results often depend on implicit assumptions, which should be articulated.
- The authors should reflect on the factors that influence the performance of the approach. For example, a facial recognition algorithm may perform poorly when image resolution is low or images are taken in low lighting. Or a speech-to-text system might not be used reliably to provide closed captions for online lectures because it fails to handle technical jargon.
- The authors should discuss the computational efficiency of the proposed algorithms and how they scale with dataset size.
- If applicable, the authors should discuss possible limitations of their approach to address problems of privacy and fairness.
- While the authors might fear that complete honesty about limitations might be used by reviewers as grounds for rejection, a worse outcome might be that reviewers discover limitations that aren't acknowledged in the paper. The authors should use their best judgment and recognize that individual actions in favor of transparency play an important role in developing norms that preserve the integrity of the community. Reviewers will be specifically instructed to not penalize honesty concerning limitations.

3. **Theory assumptions and proofs**

Question: For each theoretical result, does the paper provide the full set of assumptions and a complete (and correct) proof?

Answer: [NA]

Justification: Paper does not include theoretical results.

Guidelines:

- The answer NA means that the paper does not include theoretical results.
- All the theorems, formulas, and proofs in the paper should be numbered and cross-referenced.
- All assumptions should be clearly stated or referenced in the statement of any theorems.
- The proofs can either appear in the main paper or the supplemental material, but if they appear in the supplemental material, the authors are encouraged to provide a short proof sketch to provide intuition.
- Inversely, any informal proof provided in the core of the paper should be complemented by formal proofs provided in appendix or supplemental material.
- Theorems and Lemmas that the proof relies upon should be properly referenced.

4. **Experimental result reproducibility**

Question: Does the paper fully disclose all the information needed to reproduce the main experimental results of the paper to the extent that it affects the main claims and/or conclusions of the paper (regardless of whether the code and data are provided or not)?

Answer: [Yes]

Justification: This paper includes references to all policies used, dVRK setup, and all supplies needed to run the experiment. Details on the setup can be found in the methods and experiments section. Dataset and code are available with the submission.

Guidelines:

- The answer NA means that the paper does not include experiments.
- If the paper includes experiments, a No answer to this question will not be perceived well by the reviewers: Making the paper reproducible is important, regardless of whether the code and data are provided or not.
- If the contribution is a dataset and/or model, the authors should describe the steps taken to make their results reproducible or verifiable.
- Depending on the contribution, reproducibility can be accomplished in various ways. For example, if the contribution is a novel architecture, describing the architecture fully might suffice, or if the contribution is a specific model and empirical evaluation, it may be necessary to either make it possible for others to replicate the model with the same dataset, or provide access to the model. In general. releasing code and data is often one good way to accomplish this, but reproducibility can also be provided via detailed instructions for how to replicate the results, access to a hosted model (e.g., in the case of a large language model), releasing of a model checkpoint, or other means that are appropriate to the research performed.
- While NeurIPS does not require releasing code, the conference does require all submissions to provide some reasonable avenue for reproducibility, which may depend on the nature of the contribution. For example
  (a) If the contribution is primarily a new algorithm, the paper should make it clear how to reproduce that algorithm.
  (b) If the contribution is primarily a new model architecture, the paper should describe the architecture clearly and fully.
  (c) If the contribution is a new model (e.g., a large language model), then there should either be a way to access this model for reproducing the results or a way to reproduce the model (e.g., with an open-source dataset or instructions for how to construct the dataset).
  (d) We recognize that reproducibility may be tricky in some cases, in which case authors are welcome to describe the particular way they provide for reproducibility. In the case of closed-source models, it may be that access to the model is limited in some way (e.g., to registered users), but it should be possible for other researchers to have some path to reproducing or verifying the results.

5. **Open access to data and code**

Question: Does the paper provide open access to the data and code, with sufficient instructions to faithfully reproduce the main experimental results, as described in supplemental material?

Answer: [Yes]

Justification: The methods section links to the dataset and code along with descriptions of the robotic setup.

Guidelines:

- The answer NA means that paper does not include experiments requiring code.
- Please see the NeurIPS code and data submission guidelines (`https://nips.cc/public/guides/CodeSubmissionPolicy`) for more details.
- While we encourage the release of code and data, we understand that this might not be possible, so "No" is an acceptable answer. Papers cannot be rejected simply for not including code, unless this is central to the contribution (e.g., for a new open-source benchmark).
- The instructions should contain the exact command and environment needed to run to reproduce the results. See the NeurIPS code and data submission guidelines (`https://nips.cc/public/guides/CodeSubmissionPolicy`) for more details.
- The authors should provide instructions on data access and preparation, including how to access the raw data, preprocessed data, intermediate data, and generated data, etc.
- The authors should provide scripts to reproduce all experimental results for the new proposed method and baselines. If only a subset of experiments are reproducible, they should state which ones are omitted from the script and why.
- At submission time, to preserve anonymity, the authors should release anonymized versions (if applicable).
- Providing as much information as possible in supplemental material (appended to the paper) is recommended, but including URLs to data and code is permitted.

6. **Experimental setting/details**

Question: Does the paper specify all the training and test details (e.g., data splits, hyperparameters, how they were chosen, type of optimizer, etc.) necessary to understand the results?

Answer: [Yes]

Justification: Details on model setup, training, and hyperparameters can be found in the appendix.

Guidelines:

- The answer NA means that the paper does not include experiments.
- The experimental setting should be presented in the core of the paper to a level of detail that is necessary to appreciate the results and make sense of them.
- The full details can be provided either with the code, in appendix, or as supplemental material.

7. **Experiment statistical significance**

Question: Does the paper report error bars suitably and correctly defined or other appropriate information about the statistical significance of the experiments?

Answer: [Yes]

Justification: Associated statistical methods are described in the evaluation and results section. Averages and standard deviations are used to show how current models perform on our benchmark.

Guidelines:

- The answer NA means that the paper does not include experiments.
- The authors should answer "Yes" if the results are accompanied by error bars, confidence intervals, or statistical significance tests, at least for the experiments that support the main claims of the paper.
- The factors of variability that the error bars are capturing should be clearly stated (for example, train/test split, initialization, random drawing of some parameter, or overall run with given experimental conditions).
- The method for calculating the error bars should be explained (closed form formula, call to a library function, bootstrap, etc.)
- The assumptions made should be given (e.g., Normally distributed errors).
- It should be clear whether the error bar is the standard deviation or the standard error of the mean.
- It is OK to report 1-sigma error bars, but one should state it. The authors should preferably report a 2-sigma error bar than state that they have a 96% CI, if the hypothesis of Normality of errors is not verified.
- For asymmetric distributions, the authors should be careful not to show in tables or figures symmetric error bars that would yield results that are out of range (e.g. negative error rates).

- If error bars are reported in tables or plots, The authors should explain in the text how they were calculated and reference the corresponding figures or tables in the text.

8. **Experiments compute resources**

Question: For each experiment, does the paper provide sufficient information on the computer resources (type of compute workers, memory, time of execution) needed to reproduce the experiments?

Answer: [Yes]

Justification: Compute resources are described in the Methods section with additional training details in the Technical Appendices section.

Guidelines:

- The answer NA means that the paper does not include experiments.
- The paper should indicate the type of compute workers CPU or GPU, internal cluster, or cloud provider, including relevant memory and storage.
- The paper should provide the amount of compute required for each of the individual experimental runs as well as estimate the total compute.
- The paper should disclose whether the full research project required more compute than the experiments reported in the paper (e.g., preliminary or failed experiments that didn't make it into the paper).

9. **Code of ethics**

Question: Does the research conducted in the paper conform, in every respect, with the NeurIPS Code of Ethics `https://neurips.cc/public/EthicsGuidelines`?

Answer: [Yes]

Justification: Our work complies with the NeurIPS Code of Ethics.

Guidelines:

- The answer NA means that the authors have not reviewed the NeurIPS Code of Ethics.
- If the authors answer No, they should explain the special circumstances that require a deviation from the Code of Ethics.
- The authors should make sure to preserve anonymity (e.g., if there is a special consideration due to laws or regulations in their jurisdiction).

10. **Broader impacts**

Question: Does the paper discuss both potential positive societal impacts and negative societal impacts of the work performed?

Answer: [Yes]

Justification: Societal impacts are discussed in the societal impacts section.

Guidelines:

- The answer NA means that there is no societal impact of the work performed.
- If the authors answer NA or No, they should explain why their work has no societal impact or why the paper does not address societal impact.
- Examples of negative societal impacts include potential malicious or unintended uses (e.g., disinformation, generating fake profiles, surveillance), fairness considerations (e.g., deployment of technologies that could make decisions that unfairly impact specific groups), privacy considerations, and security considerations.
- The conference expects that many papers will be foundational research and not tied to particular applications, let alone deployments. However, if there is a direct path to any negative applications, the authors should point it out. For example, it is legitimate to point out that an improvement in the quality of generative models could be used to generate deepfakes for disinformation. On the other hand, it is not needed to point out that a generic algorithm for optimizing neural networks could enable people to train models that generate Deepfakes faster.
- The authors should consider possible harms that could arise when the technology is being used as intended and functioning correctly, harms that could arise when the technology is being used as intended but gives incorrect results, and harms following from (intentional or unintentional) misuse of the technology.
- If there are negative societal impacts, the authors could also discuss possible mitigation strategies (e.g., gated release of models, providing defenses in addition to attacks, mechanisms for monitoring misuse, mechanisms to monitor how a system learns from feedback over time, improving the efficiency and accessibility of ML).

11. **Safeguards**

Question: Does the paper describe safeguards that have been put in place for responsible release of data or models that have a high risk for misuse (e.g., pretrained language models, image generators, or scraped datasets)?

Answer: [NA]

Justification: The datasets and models in this work will only interoperate with the dVRK system. This platform is prohibited from human use by the Intuitive Foundation that issues these developer kits. Additionally, the related DaVinci system is a highly regulated medical device whose use is governed by the FDA.

Guidelines:

- The answer NA means that the paper poses no such risks.
- Released models that have a high risk for misuse or dual-use should be released with necessary safeguards to allow for controlled use of the model, for example by requiring that users adhere to usage guidelines or restrictions to access the model or implementing safety filters.
- Datasets that have been scraped from the Internet could pose safety risks. The authors should describe how they avoided releasing unsafe images.
- We recognize that providing effective safeguards is challenging, and many papers do not require this, but we encourage authors to take this into account and make a best faith effort.

12. **Licenses for existing assets**

Question: Are the creators or original owners of assets (e.g., code, data, models), used in the paper, properly credited and are the license and terms of use explicitly mentioned and properly respected?

Answer: [Yes]

Justification: External model assets, such as OpenVLA-OFT, $\pi_0$, and GR00T N1 are properly credited and used responsibly under their MIT, Apache 2.0, and Apach 2.0 licenses, respectively. Remaining assets covered in this paper are owned by the authors.

Guidelines:

- The answer NA means that the paper does not use existing assets.
- The authors should cite the original paper that produced the code package or dataset.
- The authors should state which version of the asset is used and, if possible, include a URL.
- The name of the license (e.g., CC-BY 4.0) should be included for each asset.
- For scraped data from a particular source (e.g., website), the copyright and terms of service of that source should be provided.
- If assets are released, the license, copyright information, and terms of use in the package should be provided. For popular datasets, `paperswithcode.com/datasets` has curated licenses for some datasets. Their licensing guide can help determine the license of a dataset.
- For existing datasets that are re-packaged, both the original license and the license of the derived asset (if it has changed) should be provided.
- If this information is not available online, the authors are encouraged to reach out to the asset's creators.

13. **New assets**

Question: Are new assets introduced in the paper well documented and is the documentation provided alongside the assets?

Answer: [Yes]

Justification: The new dataset provided in this paper is documented in the methods section along with the HuggingFace reference.

Guidelines:

- The answer NA means that the paper does not release new assets.
- Researchers should communicate the details of the dataset/code/model as part of their submissions via structured templates. This includes details about training, license, limitations, etc.
- The paper should discuss whether and how consent was obtained from people whose asset is used.
- At submission time, remember to anonymize your assets (if applicable). You can either create an anonymized URL or include an anonymized zip file.

14. **Crowdsourcing and research with human subjects**

Question: For crowdsourcing experiments and research with human subjects, does the paper include the full text of instructions given to participants and screenshots, if applicable, as well as details about compensation (if any)?

Answer: [NA]

Justification: Not applicable.

Guidelines:

- The answer NA means that the paper does not involve crowdsourcing nor research with human subjects.
- Including this information in the supplemental material is fine, but if the main contribution of the paper involves human subjects, then as much detail as possible should be included in the main paper.
- According to the NeurIPS Code of Ethics, workers involved in data collection, curation, or other labor should be paid at least the minimum wage in the country of the data collector.

15. **Institutional review board (IRB) approvals or equivalent for research with human subjects**

Question: Does the paper describe potential risks incurred by study participants, whether such risks were disclosed to the subjects, and whether Institutional Review Board (IRB) approvals (or an equivalent approval/review based on the requirements of your country or institution) were obtained?

Answer: [NA]

Justification: Not applicable.

Guidelines:

- The answer NA means that the paper does not involve crowdsourcing nor research with human subjects.
- Depending on the country in which research is conducted, IRB approval (or equivalent) may be required for any human subjects research. If you obtained IRB approval, you should clearly state this in the paper.
- We recognize that the procedures for this may vary significantly between institutions and locations, and we expect authors to adhere to the NeurIPS Code of Ethics and the guidelines for their institution.
- For initial submissions, do not include any information that would break anonymity (if applicable), such as the institution conducting the review.

16. **Declaration of LLM usage**

Question: Does the paper describe the usage of LLMs if it is an important, original, or non-standard component of the core methods in this research? Note that if the LLM is used only for writing, editing, or formatting purposes and does not impact the core methodology, scientific rigorousness, or originality of the research, declaration is not required.

Answer: [NA]

Justification: Not applicable.

Guidelines:

- The answer NA means that the core method development in this research does not involve LLMs as any important, original, or non-standard components.
- Please refer to our LLM policy (`https://neurips.cc/Conferences/2025/LLM`) for what should or should not be described.

