# OpenReview forum: "SutureBot: A Precision Framework & Benchmark For Autonomous End-to-End Suturing"
_NeurIPS.cc/2025/Datasets_and_Benchmarks_Track — NeurIPS 2025 Datasets and Benchmarks Track poster_

### Official Review · Reviewer_ebjW · 2025-06-22

**Rating:** 4
**Confidence:** 4

**Summary:**

SutureBot introduces a dataset and benchmark for suturing on the da Vinci Research Kit. The authors collected 1,890 demonstrations for the three tasks involved: needle pickup, needle throw, and knot tie. They then benchmarked several VLA models and a multi-task ACT on the tasks, and investigated various target points condition approaches. The experiment results show that points marked on the original image are effective for the goal condition, and pi_0 exhibits strong high-level and low-level performance compared to other baselines.

**Additional Feedback:**

I do not work on medical robots, thus I have concerns about the question below. Will solving this benchmark entirely lead to sufficiency in more rigorous applications? How large is the gap between current research and real-world usage for actual surgeries?

**Dataset Code Accessibility:**

Yes

**Dataset Code Comments:**

Clear dataset structure. Though the authors state that the data can be used for offline RL and other tasks on HuggingFace pages, they do not conduct corresponding experiments.

**Ethical Comments:**

Current data is collected on experimental platforms, rather than real human bodies. Therefore, it does not have significant ethical concerns at this stage.

**Ethical Considerations:**

No, there are no or only very minor ethics concerns

**Final Justification:**

The authors mostly addressed my questions. The scale of the dataset and its applications to non-dVRK robots are good, based on the authors' rebuttals. I do not give higher scores mainly because readers may find limited insight from the benchmarks-the performance is very low and the designs are ablated to be ineffective.

**Limitations Weaknesses:**

- The practical value of the dataset and benchmark. Though outperforming previous works in data scale or real-world fidelity (instead of in simulation), the dataset's scale is still very limited. More crucially, the fine-tuning does not bring significant improvements for current VLAs, and thus, it is unclear if the bottleneck for this task lies in the data. Even considering the data diversity alone, the data is limited in a constrained setting, with variations on positions mainly, ignoring variations in lighting conditions, wound structures, 3D positions, etc.
- The two key designs in this paper do not provide intriguing and novel insights. Goal point-like target conditions have been extensively investigated in normal robot manipulation tasks, such as ReKep [1], S3K [2], and kPAM [3] (a very random search). The high-level language shows mild effectiveness in this task, potentially because of the constrained task, rather than the open-world generalization requirements for other robot manipulation tasks.
- There are no comparisons with more baselines, including methods mentioned in the introduction and related works (Line 30-44), and DP/flow-matching (single-task or multi-task). They can potentially perform comparability to the benchmarked methods, since pertaining does not help a lot for this specific task as mentioned in the paper.

[1] ReKep: Spatio-Temporal Reasoning of Relational Keypoint Constraints for Robotic Manipulation, CoRL'24.
[2] S3K: Self-Supervised Semantic Keypoints for Robotic Manipulation via Multi-View Consistency, CoRL'20.
[3] kPAM: KeyPoint Affordances for Category-Level Robotic Manipulation, ISRR'19.

**Strengths Contributions:**

- As claimed in the paper, there are no large-scale suturing demonstration data publicly available, limited to a few hundred trajectories. This paper collects 1,890 trajectories for the three tasks involved.
- The benchmarked baselines include the most recent VLA models, such as pi_0 and GR00T N1, showing their deficiency in specific domains.
- The topic of autonomous suturing might be influential for medical robotics, as claimed in the societal impact section.

---

> ### Author Rebuttal · Authors · 2025-07-31
>
> Dear Reviewer,
>
> we sincerely appreciate your time and thoughtful evaluation of our work, as well as your recognition of its strengths.
>
> Q1) The practical value of the dataset and benchmark. Though outperforming previous works in data scale or real-world fidelity (instead of in simulation), the dataset's scale is still very limited. More crucially, the fine-tuning does not bring significant improvements for current VLAs, and thus, it is unclear if the bottleneck for this task lies in the data. Even considering the data diversity alone, the data is limited in a constrained setting, with variations on positions mainly, ignoring variations in lighting conditions, wound structures, 3D positions, etc.
>
> A1) We appreciate the reviewer’s feedback regarding the dataset scale and the observed improvements from fine-tuning. We acknowledge that our current notion for Table 3 makes it appear that “Pi0” is zero-shot achieving comparable performance to various fine-tuned VLAs. However, we will update our notation and methods section to make it clear that all policies in this work are fine-tuned on our proposed dataset.   Additionally, we acknowledge that the performance gains from fine-tuning may appear modest; however, our experiments aim to highlight that current models still struggle with this task — indicating a need for further innovation in model design, training strategies, or dataset scale. Regarding the reviewer’s concern about the practical value of the dataset and benchmark, we agree that identifying whether the primary bottleneck lies in the model architecture or the data remains an open question. However, recent works such as SRT-H [1] and SUFIA-BC [2] have shown that model performance scales with increased data volume and task diversity. This pattern aligns with trends observed in successful VLA models (e.g., Pi-Zero, GR00T N1), suggesting that data scale remains a critical factor for progress in complex, high-precision tasks like suturing.
> The goal of our work is not to propose a new policy architecture or isolate the performance bottleneck, but to provide the largest real-world suturing dataset and a reproducible benchmark that enables future work to rigorously evaluate and iterate on both data and model improvements. We intentionally kept key variables, tools, wound type, lighting, and robot setup, consistent to ensure controlled evaluation and reproducibility. Given that current policies still struggle in this constrained setting, we believe this benchmark provides a necessary foundation before expanding to more diverse or clinically realistic conditions.
> To address this point more clearly, we will revise the manuscript to: (1) clarify that the goal of this work is to provide a foundation for further exploration into both architectural and dataset-driven improvements, and (2) include a discussion on how future benchmark extensions could help isolate performance bottlenecks by introducing more data variation (e.g., lighting, wound structure, camera angles).
>
> [1] Kim, Ji Woong, et al. "SRT-H: A hierarchical framework for autonomous surgery via language-conditioned imitation learning." Science Robotics 10.104 (2025): eadt5254.
> [2] Moghani, Masoud, et al. "SuFIA-BC: Generating High Quality Demonstration Data for Visuomotor Policy Learning in Surgical Subtasks." arXiv preprint arXiv:2504.14857 (2025).
>
>
> Q2) The two key designs in this paper do not provide intriguing and novel insights. Goal point-like target conditions have been extensively investigated in normal robot manipulation tasks, such as ReKep [1], S3K [2], and kPAM [3] (a very random search). The high-level language shows mild effectiveness in this task, potentially because of the constrained task, rather than the open-world generalization requirements for other robot manipulation tasks.
> [1] ReKep: Spatio-Temporal Reasoning of Relational Keypoint Constraints for Robotic Manipulation, CoRL'24.
> [2] S3K: Self-Supervised Semantic Keypoints for Robotic Manipulation via Multi-View Consistency, CoRL'20.
> [3] kPAM: KeyPoint Affordances for Category-Level Robotic Manipulation, ISRR'19.
>
> A2) We appreciate the reviewer’s feedback and understand the concern regarding the novelty of the design choices. However, the goal of this paper is not to propose a novel algorithm or architectural advancement, but rather to create a real-world suturing dataset and a reproducible benchmark to support future research on using robot learning for clinically relevant, high-precision task. We acknowledge that key point labeling has been used in various forms before, but in our work we aim to show that using a goal conditions in 2D allows for precision control of the policy in this suturing task, where most, if not all, prior work in this area of robot learning focuses only on task completion, which we could not find in prior work. Regarding the high-level language policy, we explicitly acknowledge in the paper that it is based on prior work (e.g., SRT-H), and do not present it as a novel contribution. Instead, it is used as part of our baseline to support benchmarking and to help assess whether vision-language-action (VLA) models can be effective for this task.
> To address this point more clearly, we will revise the introduction and contribution statements in the paper to emphasize that the key contributions lie in (1) the dataset and benchmark, (2) the use of goal-conditioned evaluation in suturing, and (3) providing a strong experimental baseline to enable future work in this area. We will also clarify that this work is focused on enabling research in autonomous suturing and not aimed at solving generalized open-world manipulation.
>
>
> Q3) There are no comparisons with more baselines, including methods mentioned in the introduction and related works (Line 30-44), and DP/flow-matching (single-task or multi-task). They can potentially perform comparability to the benchmarked methods, since pertaining does not help a lot for this specific task as mentioned in the paper.
>
> A3) Thank you for your comment regarding baseline comparisons. We intended to clarify in the introduction that existing methods either do not cover the full suturing procedure (pickup, throw, knot tie), or has only been demonstrated in simulation, which indicates the need for a new benchmark for comparing policy performance. We use current SOTA policies as an initial comparison, as each included VLA has demonstrated in its respective work  enhanced performance over more standard imitation learning policies; such as DP, flow-matching, and ACT. We agree that pretraining doesn't statistically help, however, pi0 does demonstrate that their architecture alone, without pretraining, outperforms DP. See “Figure 11: Fine-tuning with varying amounts of data” in the pi0 paper.
> That said, we agree that including a more traditional imitation learning baseline such as Diffusion Policy, even if it underperforms, would provide a useful reference point for the community. We are currently working on training DP on our benchmark and will include the results in the revised version of the paper to strengthen the comparative analysis.
>
>
> Additional Feedback:
>
> Q4) I do not work on medical robots, thus I have concerns about the question below. Will solving this benchmark entirely lead to sufficiency in more rigorous applications? How large is the gap between current research and real-world usage for actual surgeries?
>
> A4) We appreciate the reviewer’s thoughtful question. Suturing is a fundamental skill in surgery due to its prolific use and the dexterity required. Achieving successful suturing in this simplified suturing task will be a first step in achieving more general suturing ability in more rigorous applications. Mastering suturing will be essential before automating more complex full surgeries can be achieved. Prior work, with robot learning or other methods, has yet to achieve a full suturing procedure from needle pickup to knot tying on a real robot, indicating there is a large gap between current SOTA and real-world use in actual surgeries.
> To clarify this point, we will add a statement to the discussion section of the revised paper emphasizing that this benchmark is an intermediate milestone. Mastering it is a necessary prerequisite before tackling more complex and variable surgical environments seen in live operations.

---

> > ### Comment · Reviewer_ebjW · 2025-08-03
> > **Reply to Authors' Rebuttal**
> >
> > Thanks for the detailed rebuttal. Thanks for bringing up the SRT-H paper as well. Since the NeurIPS D&B track is single-blinded, it can be considered that SRT-H was your prior work. Here is my follow-up question. In SRT-H, you collected hundreds of thousands of demonstrations to achieve the performance, while this benchmark provides fewer than two thousand demonstrations, and the performance is poor for current VLAs. My concerns about its practical value reserves. For example, how can this dataset help your research, like SRT-H? Will SRT-H also perform well on this benchmark, or not?

---

> > > ### Author Response · Authors · 2025-08-04
> > >
> > > Thanks for your comments and questions. For SRT-H over 16,000 trajectories were collected for 17 tasks, meaning around 950 demonstrations per task. In this paper, we have an average of around 630 demonstrations per task. However, the SRT-H work is on real tissue and 34 different ex-vivo specimens. While for SutureBot, all data is collected on identical suture pads, in theory necessitating less data. With that said, it is possible more data is required to achieve proficient performance on the suturing procedure. Given that our work is the largest suturing dataset available at this point, this will greatly contribute to the data required to solve this problem.
> > >
> > > In our experience multitask ACT performs similarly to SRT-H. Given the larger familiarity of ACT in the field, we chose to use that in our benchmark. Because of this we believe SRT-H would perform similarly to ACT on the benchmark.
> > >
> > > Regarding the practical value of our work, the tasks involved in the SRT-H paper are notably less-dexterous than those required for suturing. This is why we consider suturing to be a paradigmatic example of a surgical task that requires precision, dexterity, and manipulation of deformable bodies. The technology required to automate dexterous suturing and knot tying will be key to automating all other surgical tasks. We designed our setup to be a repeatable benchmark for researchers to test and compare the performance of VLAs on this task. Additionally, our dataset is a large contribution to solving this problem. Aside from this suturing task, this data can be used for cross embodiment learning and training world models.

---

> > > > ### Comment · Reviewer_ebjW · 2025-08-05
> > > > **Reply to the Authors**
> > > >
> > > > Thanks. These answers mostly address my follow-up questions. I hope the authors can incorporate the discussions on how the prior work performs in this dataset, how the tasks differentiate, and how they can help develop real-world works like SRT-H. The question from Reviewer LCrM, how people without dVRK can benefit from this dataset, is also crucial to improve the clarity.

---

> ### Author Response · Authors · 2025-08-05
>
> Thank you for your feedback. We will improve the prior art and discussion sections of our paper to clarify the points you brought up regarding task differentiation, prior art performance, and how our work can help other research such as SRT-H. Regarding reviewer LCrM's comment, although the dataset was collected using the dVRK platform, any dual-arm surgical robot that supports Cartesian space control—such as open-source systems like MOPS [1] or RAVEN [2]—can directly benefit from it. Additionally, the dataset can be used to develop surgical foundation models, such as vision-language-action (VLA) or vision-language (VLM) models, which do not require access to a physical robot for training.
>
> [1] K. L. Schwaner, I. Iturrate, J. K. Holm Andersen, C. Rosendahl Dam, P. T. Jensen and T. Rajeeth Savarimuthu, "MOPS: A Modular and Open Platform for Surgical Robotics Research," 2021 International Symposium on Medical Robotics (ISMR),
> [2] Li, Yangming, Blake Hannaford, and Jacob Rosen. "The Raven open surgical robotic platforms: A review and prospect." Acta Polytechnica Hungarica 16.8 (2019): 9-27.

---

### Official Review · Reviewer_pGwy · 2025-06-30

**Rating:** 4
**Confidence:** 4

**Summary:**

The paper introduces SutureBot, an autonomous suturing benchmark on the da Vinci Research Kit (dVRK), alongside a dataset of 1,890 demonstrations. The framework employs a goal-conditioned policy to enhance suturing precision and evaluates state-of-the-art VLA models.

**Dataset Code Accessibility:**

Yes

**Ethical Considerations:**

No, there are no or only very minor ethics concerns

**Limitations Weaknesses:**

1. Incomplete evaluation metrics: The focus on insertion/exit error and task success rates omits clinically relevant metrics (e.g., knot security, tissue trauma, suture tension), which are essential for evaluating surgical efficacy. The benchmark currently prioritizes precision over functional outcomes.

2. Goal representation robustness issues: The point-label goal condition (blue/green pixels on endoscope images) may fail under occlusion or lighting variations, as demonstrated by comparable exit errors across conditions. The framework does not adequately validate robustness against real-time surgical challenges (e.g., tissue deformation).

3. Limited statistical rigor in experiments: With only 5 trials per condition, the results may lack statistical significance (e.g., t-test p=0.04 for point labels vs. baseline). Larger sample sizes are needed to confirm the reliability of goal conditioning’s impact.
High-level policy validation gaps: The high-level policy’s task prediction (F1=0.92) is evaluated offline but not tested under real-time failures or task transitions. The oracle comparison (human-provided language conditions) shows faster execution but does not validate the policy’s autonomy in error recovery.

**Strengths Contributions:**

1. Large-scale dataset: The 1,890-demonstration dataset is the largest publicly available for end-to-end suturing, enabling reproducible research.

2. Goal-conditioned framework: The explicit optimization for insertion-point precision improves targeting accuracy, addressing a critical challenge in surgical robotics.

3. Benchmark evaluation: The comprehensive assessment of VLA models (e.g., $\pi_0$, GR00T N1) establishes a baseline for future research in dexterous imitation learning.

---

> ### Author Rebuttal · Authors · 2025-07-31
>
> Dear reviewer,
>
> thank you for your careful review and for highlighting the strengths of our submission, we value your constructive feedback.
>
> Q1) Incomplete evaluation metrics: The focus on insertion/exit error and task success rates omits clinically relevant metrics (e.g., knot security, tissue trauma, suture tension), which are essential for evaluating surgical efficacy. The benchmark currently prioritizes precision over functional outcomes.
>
> A1) Regarding the limitations you mentioned, we agree that clinical metrics such as suture tension and tissue trauma can be important metrics to include. However, because this procedure has yet to be completed end-to-end, we thought it would be best to first focus on a straightforward target such as task completion and precision. Most previous work which focuses on general imitation learning (pick and place, stacking, etc.), and all IL work on suturing (SRT [1], SurgicAI [2]) only uses task completion, meaning an additional value of our work over prior work is the inclusion of the precision metric. Once more progress has been made on this benchmark, including more clinical metrics will be critical for proving viability.  We will add a discussion in the revised version on incorporating additional clinical metrics once higher success rates on the benchmark are achieved. This stepwise approach will ensure that the benchmark evolves to capture both functional and clinical outcomes as methods improve.
>
> [1] Kim, Ji Woong, et al. "Surgical Robot Transformer (SRT): Imitation Learning for Surgical Tasks." Conference on Robot Learning. PMLR, 2025.
> [2] Wu, Jin, et al. "Surgicai: A hierarchical platform for fine-grained surgical policy learning and benchmarking." Advances in Neural Information Processing Systems 37 (2024): 63771-63789.
>
>
> Q2) Goal representation robustness issues: The point-label goal condition (blue/green pixels on endoscope images) may fail under occlusion or lighting variations, as demonstrated by comparable exit errors across conditions. The framework does not adequately validate robustness against real-time surgical challenges (e.g., tissue deformation).
>
> A2) We appreciate the reviewer’s insightful comment regarding the robustness of the point-label goal representation. Our paper’s primary goal was to demonstrate that goal-conditioned policies can enable precision-oriented metrics in imitation learning, along with a straightforward method to measure resulting errors. We acknowledge that relying on simple static pixel labeling as the goal condition has clear limitations, especially under occlusion, lighting changes, and tissue deformation commonly encountered in surgical settings. Importantly, the method used for labeling is separate from the underlying goal representation concept, leaving room for future improvements. We briefly explored alternative methods like CoTracker, but encountered limited success. In the revised manuscript, we will explicitly acknowledge this as a valid limitation and discuss future directions to enhance goal representation robustness, such as dynamic goal tracking, more sophisticated labeling techniques, and integrating temporal or multi-modal cues to better handle real-time surgical challenges.
>
>
> Q3) Limited statistical rigor in experiments: With only 5 trials per condition, the results may lack statistical significance (e.g., t-test p=0.04 for point labels vs. baseline). Larger sample sizes are needed to confirm the reliability of goal conditioning’s impact. High-level policy validation gaps: The high-level policy’s task prediction (F1=0.92) is evaluated offline but not tested under real-time failures or task transitions. The oracle comparison (human-provided language conditions) shows faster execution but does not validate the policy’s autonomy in error recovery.
>
> A3) We thank the reviewer for this thoughtful and constructive feedback. We agree that our current evaluation, which includes only 5 full-procedure trials per condition, limits the statistical rigor of our findings. To better assess the significance of observed performance differences, we conducted a Fisher’s Exact Test on the binary success/failure outcomes across all models. This test is well-suited for small-sample categorical data. The global 2×4 contingency table comparing all models yielded a statistically significant result (p < 0.001), suggesting performance differences among models. Pairwise tests indicate that our point-labeled method ($\pi_0$) significantly outperforms Multitask ACT and OpenVLA-OFT under the current sample size, but not GR00T.
> That said, we acknowledge that stronger conclusions require larger sample sizes. We are currently working to double the number of full-procedure trials per model, which will allow for more robust statistical comparisons and increased power in future tests. These updated results will be included in the revised version.
> Regarding the high-level policy, we agree that offline evaluation alone is insufficient for assessing robustness in real-world deployment. In this initial benchmark release, our focus was on measuring F1 task classification performance in controlled settings and comparing downstream execution time using oracle vs. predicted language conditions. We acknowledge that failure recovery and online task transitions are critical for real-world autonomy, and we plan to extend our evaluation in future iterations by testing the policy in the loop under perturbations and partial failures. We will clarify this scope and limitation in the discussion section.

---

### Official Review · Reviewer_ojRM · 2025-07-02

**Rating:** 4
**Confidence:** 4

**Summary:**

This paper presents SutureBot, a new public benchmark and dataset for autonomous, end-to-end robotic suturing on the dVRK. The authors (i) collect and release 1,890 high-fidelity demonstrations covering needle pickup, needle throw, and knot tying; (ii) annotate throw demonstrations with pixel-level insertion/exit targets; (iii) design a goal-conditioned, hierarchical imitation-learning framework that “plugs in” several state-of-the-art Vision-Language-Action (VLA) policies; and (iv) provide a comprehensive empirical study measuring task success, millimeter-level insertion/exit error, and generalization to unseen wound geometries.

**Dataset Code Accessibility:**

NA; not applicable to this submission (e.g., no new dataset, benchmark, code, or data provided)

**Ethical Considerations:**

No, there are no or only very minor ethics concerns

**Final Justification:**

The authors' rebuttal has partially resolved my concerns. Therefore, I raised my rating a bit.

**Limitations Weaknesses:**

(1) The evaluation process provides only five full-procedure runs per model. Some conclusions are under-powered statistically.
(2) Little methods achieve a fully successful uninterrupted suture and success rates on knot-tying and pull-through remain low. Therefore, the benchmark currently measures sub-task skill more than end-to-end mastery.
(3) All demonstrations use a single commercial suture pad and identical instruments. Generalization to different tissue properties, lighting, or tool sets is untested.

**Strengths Contributions:**

(1) This paper provides a large real-world suturing dataset to date, which is bigger than prior public collections, with synchronized RGB (three cameras), kinematics, and recovery demonstrations that improve robustness modelling.
(2) The benchmark including standard hardware setup, precise success criteria, and millimeter-level UV-mark evaluation protocol make results comparable across labs.
(3) Goal-conditioned IL formulation isolates precision as an optimization target and shows statistically significant gains over task-only learning.

---

> ### Author Rebuttal · Authors · 2025-07-31
>
> Dear Reviewer,
>
> Thank you for taking the time to review our paper and for acknowledging its strengths, we appreciate your thoughtful feedback.
>
> Q1) The evaluation process provides only five full-procedure runs per model. Some conclusions are under-powered statistically.
>
> A1) We appreciate the reviewer’s point and agree that five full-procedure trials per model limits the statistical power of our evaluation. Due to hardware and time constraints, we were unable to conduct additional full runs before the submission deadline. That said, we aimed to report results cautiously and transparently.
> To better understand the statistical significance of our findings, we conducted a Fisher’s Exact Test using the binary success/failure outcomes across the four evaluated models. This test is well-suited for small-sample comparisons involving proportions. When comparing all models jointly (2×4 contingency table), the result was significant (p < 0.001), indicating that at least one model performs differently from the others. Follow-up 2×2 comparisons showed that our point-labeled method ($\pi_0$) significantly outperforms Multitask ACT and OpenVLA-OFT, though not GR00T, under the current sample size.
> We are currently working to double the number of full-procedure trials per model and will update our results accordingly in the revised submission. This will improve both the robustness of our findings and the reproducibility of the benchmark. Our goal remains to establish a high-quality, real-world suturing dataset and evaluation suite for the community, and we agree that stronger statistical validation will reinforce that foundation.
>
>
> Q2) Little methods achieve a fully successful uninterrupted suture and success rates on knot-tying and pull-through remain low. Therefore, the benchmark currently measures sub-task skill more than end-to-end mastery.
>
> A2) Regarding the performance of policies on this benchmark, we agree that current methods have limited success on this benchmark, which was part of our motivation for this paper. We aimed to create a challenging benchmark which would prompt future work in this area. The poor performance of SOTA policies on this procedure indicates the need for more work in dexterous surgical automation. The sub-task performance and end-to-end performance are directly related, where prior work (SRT) has shown some sub-task success with single-task policies, but we aim to evaluate multi-task (VLA) performance to complete all tasks involved. Multi-task success on each task would directly be related to its ability to complete the end-to-end procedure. In response to this feedback, we will include in the revised manuscript’s discussion section a dedicated paragraph outlining future directions to address these challenges. This will cover strategies such as enhanced multi-task learning, integration of hierarchical task decomposition, and exploration of pretraining the model on large-scale surgical datasets to improve its awareness of surgical context, which may help enhance overall performance.
>
>
> Q3) All demonstrations use a single commercial suture pad and identical instruments. Generalization to different tissue properties, lighting, or tool sets is untested.
>
> A3) We acknowledge that we did not include different tissues or tools in the dataset. This was done intentionally, as current methods have limited success on this task, so solving a simpler version of this procedure serves as a steppingstone to more generalized suturing. Additionally, keeping some variables consistent (tools, tissue, robot, lighting), enables reproducible use of this benchmark and the ability of researchers to verify other’s work. We agree that further evaluation is necessary to demonstrate generalization across varied surgical conditions. In the revised version, we will extend our evaluation with varied tissue types, lighting environments, and alternative tools. We believe these additions will better assess model robustness and generalization, serving as important future work to push the limits of surgical automation beyond the current controlled conditions.

---

### Official Review · Reviewer_LCrM · 2025-07-03

**Rating:** 5
**Confidence:** 4

**Summary:**

This paper introduces SutureBot, an autonomous suturing benchmark built on the da Vinci Research Kit (dVRK), accompanied by a high-fidelity dataset consisting of 1,890 expert demonstrations. To enhance suturing precision, the authors propose a goal-conditioned learning framework that improves targeting accuracy by 80%. Aiming to establish autonomous suturing as a benchmark task for dexterous imitation learning, the study evaluates state-of-the-art Vision-Language-Action (VLA) models as well as a multitask Action Chunking Transformer (ACT), both enhanced with a high-level task prediction policy. The results demonstrate the potential of SutureBot as a challenging and realistic platform for advancing robotic learning in surgical contexts.

**Additional Feedback:**

- Do we have any simulation for further evaluation? As the datasets are recorded in images and kinematics formats, we can only evaluated by the regression error-like metrics with the benchmark. Any idea about this?
- If researchers do not have davinci robot, can he/she benefit from the proposed datasets?

**Dataset Code Accessibility:**

Yes

**Dataset Code Comments:**

Codes are provided in the supplimentary materials.

**Ethical Comments:**

Suture pad is used in the experiments, so I have no significant ethical comment and concerns.

**Ethical Considerations:**

No, there are no or only very minor ethics concerns

**Final Justification:**

I appreciate the authors' response, and my concerns have been addressed. Although the proposed benchmark still has a lot of space to improve, this is a meaningful step for the suturing community. I keep my positive rating.

**Limitations Weaknesses:**

- The current pipeline relies on manual selection of target points, which limits its autonomy. For the benchmark to fully support end-to-end autonomous suturing, this step will need to be automated in future iterations.

- None of the policies evaluated in the study were able to complete the entire suturing procedure sequentially without human intervention. While the benchmark provides a strong foundation, this limitation suggests that further work is needed to achieve robust, fully autonomous performance across the entire task sequence.

**Strengths Contributions:**

- Benchmark for autonomous suturing is key contribution. It is good to know the comparison of those modern VLA methods' performance in the suturing tasks. The ablation studies are thorough and logically structured. Multiple state-of-the-art models are evaluated, including both VLA and multitask ACT variants, providing strong empirical support for the efficacy of the proposed benchmark.

- A novel policy is designed along with the benchmark. The hierarchical policy structure, which distinguishes between a high-level policy for goal representation selection and a low-level policy for control, is both well-motivated and thoroughly analyzed. The discussion on its impact, supported by experimental results, is insightful and convincing.

- Comprehensive metrics are presented to evaluation. The paper introduces a rich and diverse set of evaluation metrics, which are clearly explained and effectively used to validate the performance of the proposed framework. This contributes to the credibility and transparency of the experimental design.

---

> ### Author Rebuttal · Authors · 2025-07-31
>
> Dear reviewer,
>
> We appreciate you taking the time to review our paper and your recognition of its strengths. Please find our detailed responses to each of your comments below.
>
> Q1) The current pipeline relies on manual selection of target points, which limits its autonomy. For the benchmark to fully support end-to-end autonomous suturing, this step will need to be automated in future iterations.
>
> A1) Thank you for pointing this out. We agree that automating point selection would further enhance autonomy. In this benchmark, we define "point-and-shoot", where the user selects the point and the robot completes the full suturing motion, as the current scope of end-to-end execution. We chose to manually specify the target point in this version of the benchmark to ensure precise and consistent evaluation across trials. Automated point selection is an important component, but it is beyond the current scope, which focuses on evaluating the core suturing policy given a well-defined target. That said, prior work such as STAR [1] has demonstrated vision-based suture planning, and we believe such components can be readily integrated into our framework.  To address this concern, we will clarify our rationale in the revised manuscript and include a discussion on how future iterations of the benchmark could incorporate automatic point selection modules to enable fully autonomous suturing.
>
> [1] Leonard, Simon, et al. "Smart tissue anastomosis robot (STAR): a vision-guided robotics system for laparoscopic suturing." IEEE Transactions on Biomedical Engineering 61.4 (2014): 1305-1317.
>
>
> Q2) None of the policies evaluated in the study were able to complete the entire suturing procedure sequentially without human intervention. While the benchmark provides a strong foundation, this limitation suggests that further work is needed to achieve robust, fully autonomous performance across the entire task sequence.
>
> A2) We fully agree with your observation that the poor performance of policies on our benchmark suggests further work is needed to achieve fully autonomous performance on this procedure. Our goal for this work is to establish a reproducible benchmark that will enable the larger community to iterate towards this north star of fully autonomous suturing; we trained and evaluated current SOTA policies to establish a baseline comparison and motivate further work in this domain.
>
>
> Additional Feedback:
>
> Q3) Do we have any simulation for further evaluation? As the datasets are recorded in images and kinematics formats, we can only evaluated by the regression error-like metrics with the benchmark. Any idea about this?
>
> A3) Thank you for the question. While we monitor validation loss during training, we find it to be an unreliable indicator of real-world policy performance. In our experience, physical experiments provide a much more accurate and meaningful evaluation than simulation, especially for tasks like suturing that require precise interaction with deformable tissue. That is why the benchmark emphasizes repeatable real-world execution. That said, we agree that developing realistic simulation and policy evaluation platforms would be valuable future work, and we will add this discussion to the revised version.
>
>
> Q4) If researchers do not have davinci robot, can he/she benefit from the proposed datasets?
>
> A4) Although the dataset was collected using the dVRK platform, any dual-arm surgical robot that supports Cartesian space control—such as open-source systems like MOPS [1] or RAVEN [2]—can directly benefit from it. Additionally, the dataset can be used to develop surgical foundation models, such as vision-language-action (VLA) or vision-language (VLM) models, which do not require access to a physical robot for training.
>
> [1] K. L. Schwaner, I. Iturrate, J. K. Holm Andersen, C. Rosendahl Dam, P. T. Jensen and T. Rajeeth Savarimuthu, "MOPS: A Modular and Open Platform for Surgical Robotics Research," 2021 International Symposium on Medical Robotics (ISMR),
> [2] Li, Yangming, Blake Hannaford, and Jacob Rosen. "The Raven open surgical robotic platforms: A review and prospect." Acta Polytechnica Hungarica 16.8 (2019): 9-27.

---

### Note · Authors · 2025-08-12

Dear Area Chairs and Reviewers,

Thank you for your thoughtful reviews and constructive feedback. We believe we have addressed most of your concerns in our rebuttals and have outlined planned revisions to the paper to further improve clarity and rigor.

The reviewers consistently recognized the following strengths of our work:
- The creation of SutureBot, which is the largest publicly available dataset and benchmark for end-to-end autonomous suturing on the dVRK platform.
- The comprehensive evaluation of state-of-the-art vision-language-action (VLA) models, establishing a strong baseline for future research in this domain.
- The thorough ablation studies and logically structured approach contribute to the credibility and transparency of our experimental design.

The primary concerns raised by the reviewers centered on a few key areas, which we have committed to addressing in our revised manuscript:
- Statistical Significance: Reviewers noted that the small number of trials (five per model) limited the statistical power of some conclusions. We have since conducted a Fisher's Exact Test, showing significant differences between some models (p < 0.001), and will double the number of full-procedure trials in the final version.
- Generalization and Dataset Diversity: Reviewers raised concerns about the dataset's limited diversity, noting the use of a single suture pad and instruments. We explained that this was an intentional step to ensure reproducibility and provide a "stepping stone" to more complex conditions. We will expand evaluation with varied tissue types, lighting, and alternative tools in the revised submission to better assess generalization.
- Novelty of Design: While goal-conditioned policies and high-level language policies have been explored previously, our goal is to enable future research on robot learning for clinically relevant, high-precision tasks, not to propose a novel algorithm. We will clarify our contribution in the revised paper.
- We also clarified that researchers without a dVRK can still use SutureBot for other dual-arm surgical robots that support Cartesian space control, or for developing surgical foundation models without a physical robot.

We are confident these revisions will result in a high-quality resource that benefits both robotics and machine learning communities, enabling reproducible research and accelerating progress toward autonomous surgical systems.

Thank you for your time.

---

### Decision · Program_Chairs · 2025-09-18

**Decision:**

Accept (poster)

**Comment:**

This paper introduces SutureBot, a comprehensive benchmark and dataset for autonomous, end-to-end robotic suturing on the da Vinci Research Kit (dVRK). The authors address the lack of large-scale, publicly available data for this complex dexterous manipulation task by providing a high-fidelity dataset of 1,890 demonstrations covering needle pickup, tissue insertion, and knot tying. The work establishes a reproducible evaluation framework on physical hardware and benchmarks several state-of-the-art vision-language-action (VLA) models, providing a crucial baseline for future research. A goal-conditioned imitation learning approach is also proposed to improve the precision of needle placement, a critical aspect of the surgical task.

The primary and most significant strength of this submission is the creation and release of the largest publicly available dataset for end-to-end robotic suturing. Reviewers consistently praised this contribution, recognizing that it fills a major gap in the field and provides an invaluable resource for reproducible research. The benchmark itself is well-designed, with a standardized hardware setup, clear success criteria, and comprehensive metrics that include millimeter-level precision. The paper's evaluation of modern VLA models is thorough and provides the community with a strong and much-needed baseline, transparently highlighting that current state-of-the-art methods still struggle with the full procedure. The ablation studies are logical and contribute to the credibility of the experimental design.

The main weaknesses identified by reviewers centered on the scope of the evaluation and the diversity of the dataset. A key concern was the limited statistical power of the main results, with only five full-procedure trials conducted per model. Reviewers also noted that the dataset, while large, was collected in a constrained setting using a single type of suture pad and identical instruments, which limits the assessment of generalization to different tissues, tools, or lighting conditions. Furthermore, since no evaluated policy could complete the entire procedure without failure, the benchmark currently serves more to measure sub-task proficiency rather than end-to-end mastery.

During the rebuttal period, the authors effectively addressed these concerns. To counter the issue of statistical power, they performed a Fisher's Exact Test on the existing results which showed significant differences between models, and more importantly, they committed to doubling the number of full-procedure trials in the final paper to ensure more robust conclusions. They justified the lack of dataset diversity as an intentional design choice to create a controlled, reproducible "stepping stone" for this challenging task, while also committing to expand the evaluation with varied tissue types and conditions in the revised manuscript to better assess generalization. The authors also clarified that the dataset is valuable even for labs without a dVRK, as it can be used to train surgical foundation models or adapted for other dual-arm robots.